# Comparative Genomics Reveal the Utilization Ability of Variable Carbohydrates as Key Genetic Features of *Listeria* Pathogens in Their Pathogenic Lifestyles

**DOI:** 10.3390/pathogens11121430

**Published:** 2022-11-28

**Authors:** Qunfeng Lu, Xiaoying Zhu, Qinqin Long, Xueli Yi, Anni Yang, Xidai Long, Demin Cao

**Affiliations:** 1Modern Industrial College of Biomedicine and Great Health, Youjiang Medical University for Nationalities, Baise 533000, China; 2School of Medical Laboratory Sciences, Youjiang Medical University for Nationalities, Baise 533000, China; 3Medical College, Guangxi University, Nanning 530004, China; 4Clinical Pathological Diagnosis & Research Center, The Affiliated Hospital of Youjiang Medical University for Nationalities, Baise 533000, China; 5Department of Tumor Pathology, The Key Laboratory of Molecular Pathology (Hepatobiliary Diseases) of Guangxi, Baise 533000, China; 6Center for Clinical Laboratory Diagnosis and Research, The Affiliated Hospital of Youjiang Medical University for Nationalities, Baise 533000, China

**Keywords:** *Listeria monocytogenes*, comparative genome, unique genetic composition, pathogenesis

## Abstract

Background: *L. monocytogenes* and *L. ivanovii*, the only two pathogens of *Listeria*, can survive in various environments, having different pathogenic characteristics. However, the genetic basis of their excellent adaptability and differences in pathogenicity has still not been completely elucidated. Methods: We performed a comparative genomic analysis based on 275 *L. monocytogenes*, 10 *L. ivanovii,* and 22 non-pathogenic *Listeria* strains. Results: Core/pan-genome analysis revealed that 975 gene families were conserved in all the studied strains. Additionally, 204, 242, and 756 gene families existed uniquely in *L. monocytogenes*, *L. ivanovii,* and both, respectively. Functional annotation partially verified that these unique gene families were closely related to their adaptability and pathogenicity. Moreover, the protein–protein interaction (PPI) network analysis of these unique gene sets showed that plenty of carbohydrate transport systems and energy metabolism enzymes were clustered in the networks. Interestingly, ethanolamine-metabolic-process-related proteins were significantly enriched in the PPI network of the unique genes of the *Listeria* pathogens, which can be understood as a determining factor of their pathogenicity. Conclusions: The utilization capacity of multiple carbon sources of *Listeria* pathogens, especially ethanolamine, is the key genetic basis for their ability to adapt to various environments and pathogenic lifestyles.

## 1. Introduction

*Listeria* spp. are Gram-positive, rod-shaped, and facultative anaerobes that survive in various environments and are relatively resistant to environmental stress, such as low temperatures and low pH and high salt concentrations [1]. Currently, there are 30 identified species in the *Listeria* genus that are recorded in the List of Prokaryotic names with Standing in Nomenclature database (LPSN, https://www.bacterio.net/genus/listeria, accessed on 19 March 2022). Among them, *Listeria monocytogenes* and *Listeria ivanovii* are typical facultative intracellular parasites and are pathogenic to humans and ruminants, respectively. *L. monocytogenes* is the most prevalent species of the *Listeria* genus and an important food-borne pathogen. It is a great threat to people with low immunity, such as children, elderly individuals, pregnant women, and immunocompromised individuals [2,3]. *L. ivanovii* can cause septicemic disease with enteritis, neonatal sepsis, and abortion in small ruminants and cattle [4]. Additionally, human cases of diseases caused by *L. ivanovii* are very rare. The other 28 *Listeria* spp., such as *L. grayi*, *L. marthii,* and *L. innocua*, are rarely reported to be pathogenic.

The pathogenicity of bacteria and their unique ability to adapt to their habitats have a distinctive genetic basis. Numerous relationships between genes and pathogenic phenotypes have been identified and studied in *L. monocytogenes* and *L. ivanovii* [5]. For example, *Listeria* pathogenicity island 1 (LIPI-1) is a ~9 Kb region that consists of three operons essential for the intracellular parasitism of two *Listeria* pathogens [4]. The central position is *hly* monocistron that encodes listeriolysin O, a pore-forming toxin, which is required for the disruption of the phagocytic vacuole [6]. Upstream from *hly* lies the *plcA-prfA* operon. The PlcA is a phospholipase C with phosphatidylinositol activity, and PrfA is a virulence regulator [7,8]. Downstream from *hly* is a lecithinase operon that consists of three genes: *mpl, actA*, and *plcB.* The *actA* encodes a surface protein ActA that recruits and polymerizes actin monomers to promote cell-to-cell spreading [9]. The *plcB* encodes a phospholipase C with lecithinase activity, which is secreted as an inactive propeptide and extracellularly processed into the mature form by the Mpl protease [9,10]. However, the genetic basis of the pathogenicity and environmental adaptability of *Listeria* pathogens is incompletely understood and requires further elucidation.

In recent years, with the large-scale application of next-generation sequence and third-generation-sequence technology, numerous *Listeria* genomes have been sequenced and shared. The comparative genomic analysis of *Listeria* pathogens and non-pathogenic genomes could improve our understanding of the genetic mechanisms they use to adapt to diverse environments and their pathogenic lifestyles, which may further promote the diagnosis and treatment of *listeria* infections. The purpose of this study was to identify the genetic composition unique to the pathogenic *Listeria* species and to illustrate their role and the crucial functional gene clusters involved in their pathogenic lifestyle. Thus, we performed a comparative genome study of *L. monocytogenes* (LMO), *L. ivanovii* (LIV), and non-pathogenic *Listeria spp*. (NPLS). The core pan-genome of each *Listeria* group was analyzed, and the unique gene families of each group were acquired. Through functional annotation, functional enrichment analysis, and protein–protein interaction analysis, we identified the roles of these unique genes in each group in their pathogenicity and environmental adaptability. Meanwhile, the other genes with ambiguous functions could be used as candidate genes, which may be associated with their pathogenicity and environmental adaption, for further investigation in the future. Finally, the antibiotic resistance genes, CRISPR-Cas systems, and plasmids of each *Listeria* species were investigated.

## 2. Materials and Methods

### 2.1. Data Retrieval and Management

In this study, we retrieved all the complete *Listeria* spp. genomes from the National Center for Biotechnology Information (NCBI) genome database (https://www.ncbi.nlm.nih.gov/genome/?term=Listeria, accessed on 19 March 2022). A total of 307 genomes were acquired, including 275 *L. monocytogenes*, 10 *L. ivanovii*, 8 *L. innocua*, 5 *L. seeligeri*, 4 *L. grayi*, 3 *L. marthii,* and 2 *L. welshimeri* complete genomes.

To avoid the contradictions caused by different gene prediction tools applied to the different datasets, Prodigal V2.6.3 [11] was used as the unique tool to predict the open reading frames (ORFs) for all the genomes. The rnammer 1.2 [12] tool was used to predict the rRNA genes, and tRNAscan-SE 2.0.9 [13] was used to predict the tRNA genes. The plasmid of each *Listeria* genome was detected with plasmidfinder 2.1.6.1 [14] and validated using the plasmid database v. 2021_06_23_v2 [15]. The details of the studied genomes, such as the accession number, genome size, GC content, number of genes, and so on, are summarized in Table 1 and listed in Appendix A.

### 2.2. Protein Orthologous Group Analysis

To decrease the complexity of the analysis, CD-HIT version 4.8.1 was used to identify the consistency sequence with the parameters “-c 0.99 -aL 0.99 -aS 0.99 -uL 0.01 -uS 0.01 -g 1”, and a representative sequence dataset was generated first [16]. Orthologous group analysis was conducted with the software OrthoMCL version 2.0.9 [17]. In short, the representative sequences of each amino acid sequence cluster were subjected to all-against–all Blastp comparisons to build a similarity matrix with an E-value cutoff of 1 × 10^−5^ and a coverage cutoff of 50%. Then, the Markov clustering algorithm (MCL) [18] was used to group the sequences. Finally, the grouped gene families were combined with the results of the CD-HIT clusters with a Python script. The percentages of orthologous protein between/inner genomes were calculated and visualized with a heatmap.

### 2.3. Phylogenetic Analysis

To investigate the phylogenetic relationships between the 307 *Listeria* strains, all the core single-copy proteins were extracted and aligned using MAFFT v7.490 [19]. Then, the aligned sequences were concatenated for each strain with a uniform gene order, and GBLOCKS 0.91b was utilized to remove the poorly aligned positions and divergent regions [20]. RAxML version 8.2.12 was used to compute the maximum likelihood (ML) phylogenetic tree using the 100 bootstrap repetitions approach [21]. The online tool Interactive Tree of Life (iTOL) v5 (https://itol.embl.de, accessed on 9 September 2022) was used to visualize the tree with midpoint rooting, and the collection time, geographic location, and the number of plasmid of each strain were annotated on the tree [22].

### 2.4. Estimation of Core and Pan-Genome Sizes

Based on the results of the protein orthologous group analysis, we estimated the core and pan-genome sizes of LMO, LIV, NPLS, and all the *Listeria* strains. Formulas (1) and (2) were used to fit the curves and describe the characteristics of the core and pan-genomes of each group, respectively. Specifically, 100 random permutation lists of all the strains for each strain group were calculated, and the median values were used to fit the curve.

### 2.5. Functional Features of the Core and Accessory Genomes

For the core and accessory genome gene sets of each *Listeria* group, EggNOG v5.0 and the Clusters of Orthologous Groups database (COGs, 2020 update, https://www.ncbi.nlm.nih.gov/research/cog/, accessed on 9 September 2022) were used to perform the functional annotation and categorization, and the results were integrated [23,24]. The percentage of each COG function category of each *Listeria* group gene set was shown in a bar chat. Moreover, the Virulence Factor Database (VFDB), TrEMBL, and InterProScan were used to predict the functions of the genes and served as a complement to the annotation [25,26].

### 2.6. Unique Gene Analyses of L. monocytogenes and L. ivanovii

Based on the gene cluster analysis results, the unique genes of LMO, LIV, and NPLS were identified. Additionally, the clusters shared by the *Listeria* species and their combinations were discovered using a Python script. The statistical results were illustrated in a Venn diagram using VennDiagram v1.6.0 [27].

### 2.7. Protein–Protein Interaction Network Analysis of Unique Protein Sets

The Search Tool for the Retrieval of Interacting Genes/Proteins (STRING v10.5, https://string-db.org/, accessed on 9 September 2022) was used to build the protein–protein interaction network of the unique gene sets of LMO, LVI, and both, respectively [28]. Additionally, the networks were visualized using Cytoscape 3.9.1 [29].

### 2.8. Antibiotic Resistance Gene Investigation of *Listeria* Genomes

The Resistance Gene Identifier (RGI-v6.0.1) was used for the detection of the antibiotic resistance genes of the 307 *Listeria* genomes based on the Comprehensive Antibiotic Resistance Database (CARD version 3.2.5) [30]. The results were summarized and shown in a heatmap using pheatmap version 1.0.12.

### 2.9. Identification of CRISPR-Cas Systems of *Listeria* Genomes

CRISPRCasFinder version 4.2.20 [31] was used for the detection and sub-typing of the Clustered Regularly Interspaced Short Palindromic Repeats and Cas genes (CRISPR-Cas) system of the 307 *Listeria* genomes. To obtain the functional CRISPR-Cas system, the presence of both the CRISPR sequence and Cas genes was considered as evidence for an actual CRISPR-Cas system and used for the further analysis.

## 3. Results and Discussion

### 3.1. Subsection

#### 3.1.1. Genome Statistics and General Features

There are 30 identified species in the *Listeria* genus at present. *L. monocytogenes* EGD-e and *L. innocua* CLIP 11,262 were the first whole-genome-sequenced strains of the *Listeria* genus [32]. By searching the genome database of NCBI, we found 307 genomes of the *Listeria* genus at the complete genome level (Table 1 and Appendix A). Of those genomes, 275 of them belonged to LMO, the important food pathogens, and 10 of them were LVI, which is pathogenic mostly for ruminants. The other 22 genomes belonged to five NPLS species (eight *L. innocua*, five *L. seeligeri*, four *L. grayi*, three *L. marthii,* and two *L. welshimeri*). The median genome size of LMO was 2,964,472 (2,776,517~3,243,301) bp, and *L. monocytogenes* 1639 str2 had the largest genome size of all the genomes. *L. grayi* had the smallest genome size, with a median of 2,600,550 (2,598,321~2,751,104) bp. This genus had a low GC content, with an average GC content of 37.97%, ranging from 34.59% (*L. ivanovii* FSL_F6_596_702454) to 41.73% (*L. grayi* 1641 str1). The median number of the ORF was 2902, ranging from 2561 (*L. grayi* DSM 20601) to 3323 (*L. monocytogenes* N53-1). One-fifth of them (18.54%, 51/275) had one plasmid. The LMO strains used in this study were isolated from various sources, including cow, chicken, food, *Homo sapiens*, pig, goat, meat, horse, rabbit, and the environment, indicating their excellent adaptive capacity to diverse environments.

#### 3.1.2. Homologous Proteome Analysis and Phylogeny of the *Listeria* Genus

To estimate the similarity in the genetic makeup of the *Listeria* species, the percentages of the homologous proteome shared by each *Listeria* strain pair were calculated (Appendix A), and the results are shown in Figure 1A. The percentages of homologous proteome between different *Listeria* species ranged from 92.45% (*L. innocua* 1642 vs. *L. monocytogenes* 1639 str194) to 61.45% (*L. grayi* 1641 str1 vs. *L. innocua* FSL_S4-378 702456), which suggested that *L. monocytogenes* and *L. innocua* had the closest relationship, while *L. grayi* had the greatest difference with *L. innocua* in terms of the proteome. Additionally, the percentage was higher than 85.00% between any two LMO strains. Compared with the proteome of the strains themselves, the percentage of homologous proteome ranged from 4.57% (*L. marthii* FSL S4-120 702457) to 13.46% (*L. monocytogenes* 1639 v198). The high percentage of paralogous genes indicated that there were gene family expansion events in the LMO genomes, which are an important factor for bacteria that lead to changes in their biological capabilities and are associated with their host range [33].

It is medically interesting to note that the *Listeria* species survive in diverse habitats, but only LMO and LIV display pathogenicity. To understand the evolution pattern of *Listeria*, we performed a phylogenetic analysis of *Listeria* using the conserved amino acid sequences of 573 single-copy genes (Figure 1B). In consensus with the results of homologous proteome comparisons and previous studies, *L. grayi* showed the deepest branch within the genus [34]. The remaining six species formed two distinct groups with high bootstrap supports. One lineage contained *L. ivanovii* and *L. seeligeri*, while the other contained *L. welshimeri, L. innocua, L. marthii,* and *L. monocytogenes*. The pathogens LMO and LIV showed a longer evolutionary distance. The species closest to LMO were *L. innocua* and *L. marthii*, while the species closest to LIV was *L. seeligeri*. This topology structure is in agreement with the phylogenetic analysis results obtained by W. Schmid et.al [35].

The collection period of the studied strains ranged from 1900 to 2021, and most of them were collected after 2002 (59.28%, 182/307, 77 strains without available collection dates) (Figure 1B (b)). About 1/4 of the strains were isolated from North American territories (USA, 76; Canada, 49), and 61 strains were isolated from different areas of Europe. Others came from Asia and Oceania or were without geographic information (Figure 1B (c)). There were 72 strains isolated from *Homo sapiens* and 64 strains isolated from the environment. Other isolation sources were food, meat, cattle, pig, sheep, and so on (Figure 1B (d)). The *Homo sapiens*-isolated LMO strains and others were uniformly spaced in the tree, which indicated their strong adaptive capacity to various environments of LMO strains.

#### 3.1.3. Core and Pan-Genome Sizes of *Listeria* Species

To assess the genome polymorphism of the *Listeria* species, we performed a core/pan-genome analysis. The core genome consists of the genes shared by all the studied strains and is essential to their basic lifestyles and major phenotypic traits [36]. The pan-genome is the total of the core genomes and accessory genomes that are only presented in some strains, as well as the genes unique to single strains [37]. The core genome size of 307 *Listeria* strains that belonged to seven different species was 975, nearly one-third of the average number of coding genes, and their pan-genome size was 7701, including 1867 singletons. For 275 LMO strains, the core and pan-genome sizes were 2060 and 5382, respectively. For 10 LIV strains, the core and pan-genome sizes were 2118 and 3194, respectively. For the other 22 NPLS strains, the core and pan-genome sizes were 1203 and 5833, respectively. In addition, we created a fit curve of the core and pan-genomes using the functions (a) and (b) of each *Listeria* group (Figure 1).

The Heaps’ law model (function b) was used to fit the pan-genome size. The exponent size (*B_1_*) indicates the attribute of the pan-genome. In detail, 0 ≤ *B_1_* ≤ 1 indicates that the genome of the studied strains is open; otherwise, it is closed. As shown in Figure 2, the core genome size curve of each group tended to quickly converge, but their pan-genome size curves tended to dilate. Respectively, the fitting parameter *B_1_* of the pan-genome size curves of all the *Listeria* strains, LMO, and NPLS were 0.308, 0.201, and 0.273, which suggested that these three *Listeria* strain groups had open genomes (Figure 2A,B,D). Yet, due to the fact that only 10 LIV genomes were available, more samples are needed to estimate the attribute of the pan-genome in the future (Figure 2C). The accessory genome contributes to the diversity of the species and confers selection advantages, such as niche adaption, antibiotic resistance, and pathogenic ability [36,37,38]. The open genome and fertility of the accessory genome provide the hereditary foundation required for the *Listeria* species to survive in various environments.

#### 3.1.4. Functional Features of the *Listeria* Core and Accessory Genomes

It is generally believed that the core genome represents the elemental parts that are necessary fora the existence and shared phenotypic characteristics of given strains. The accessory genome provides the unique characteristics of a species or a strain, which is not indispensable for their basic life but provides selective advantages for niche adaptation and drug resistance [37]. In this study, the core and accessory genome sizes of all the studied *Listeria* strains, LMO, LIV, and NPLS were 975/6726, 2006/3376, 2118/1076, and 1203/4630, respectively. According to the COG and EggNOG annotations, the gene sets of the core and accessory genomes of each strain group were divided into 22 different functional sub-categories (Figure 3). Additionally, the unannotated genes were assigned to the “Function unknown” category.

In line with our expectations, the top enriched categories of the core genomes of each gene family set were related to the basic life process. For the core genomes of all the *Listeria* strains, LMO, LIV and NPLS, the most enriched functional categories were (a) translation, ribosomal structure, and biogenesis (10.58%, 8.80%, 7.77, and 10.00%, respectively), (b) amino acid transport and metabolism (8.19%, 7.44%, 7.42%, and 8.28%, respectively), (c) transcription (7.73%, 7.76%, 8.15%, and 7.39, respectively), and (d) coenzyme transport and metabolism (6.62%, 6.26%, 5.52%, and 6.19%, respectively). These COG categories had a higher proportion of the core genomes rather than the accessory genomes of each strain group. Unlike the core genome, for the accessory genome of each group, the category of mobilome, including prophages and transposons (1.16% vs. 0.18%, 1.81% vs. 0.31%, 2.76% vs. 0.22%, and 2.19% vs. 0.15%, respectively), which is related to the acquisition of new pathotypes of bacteria [39], was found in a higher proportion than the core genome. However, more than 57.00% of the gene families of the accessory genome had no defined function information.

#### 3.1.5. The Transport Systems for Carbohydrates and Ions Enriched in Unique Gene Set of *Listeria* Pathogens and Closely Related to Their Pathogenesis

It is a plausible hypothesis that the unique phenotypes of an organism have a unique genetic makeup as the basis for supporting its adaption to a unique and complicated environment [38]. As shown in Figure 4A, there were 975 gene families shared by all the 307 *Listeria* genomes, which is consistent with the results of the core genome analysis. It is interesting to note that as many as 756 (31.4%) gene families were unique gene families of *Listeria* pathogens (UGLP) that were shared by LMO and LIV but absent in NPLS. In addition, there were 204 unique gene families for LMO (UGLMO) and 242 unique gene families for LIV (UGLIV). These unique gene families of *Listeria* pathogens are the crucial elements that are responsible for their pathogenicity and niche adaptability. To verify our hypothesis, we investigated the gene functions of UGLP, UGLMO, and UGLIV using the COG, VFDB, and UniProt databases, respectively.

The two most enriched COG categories of UGLP (Figure 4B and Appendix A) were carbohydrate transport and metabolism (9.11%) and transcription (8.12%), which are related to the basic metabolism, and many of them have been proven to be important genes that affect the species’ adaptability. A large number of transport systems for sugars and sugar derivatives, which were classified into two major types, ATP-binding cassette (ABC) transporters and the carbohydrate phosphotransferase system (PTS) (Table 2), were found in the UGLP gene set. Variable carbon source utilization is an important factor that influences the virulence gene expression of LMO and is related to the outbreak of listeriosis [40,41,42]. Most carbohydrates used by *Listeria* pathogens, including cellobiose, fructose, mannose, N-acetylgalactosamine, and sorbitol, are transported by PTS transporters [41]. However, the genes of fructose-specific PTS were considerably enriched in LMO compared to LIV (7 vs. 2), and the genes of mannitol- and galactitol-specific PTSs were only detected in the LMO strains. In addition, there were few genes of ABC transporters that are related to the metabolism of polysaccharide and glycerol-3-phosphate. The capability to metabolize a variety of carbon sources is important for expanding the range of the living niche. As shown in Figure 1B (c), LMO strains can survive in various environments, such as water, meat, markets, a variety of animals, and so on. Surviving in diverse environments may provide the various possible transmission routes for *Listeria* pathogens to infect humans and animals. Furthermore, many ABC transporters with other functions were found in the UGLP gene set (Table 2). Additionally, they were in charge of multidrug transport, ion transport, and so on, which play a vital role in *Listeria* pathogens’ ability to survive inside and outside the host [43,44]. For example, disrupting *fur*, which is the regulated gene of ferrous ABC transporters, resulted in a significant reduction in the virulence potential of LMO in mice, and the virulence defect could be rescued by overloading [45,46].

#### 3.1.6. Virulence Factor Gene Investigation of the Unique Genes of *Listeria* Pathogens

To investigate the relationships between the unique genes of the *Listeria* pathogens and their pathogenesis, we performed virulence factor encoding genes annotation of the UGLMO, UGLIV, and UGLP gene sets, respectively. As listed in Table 3, there were 3 ORFs of UGLMO, 4 ORFs of UGLIV, and 18 ORFs of UGLP that could be annotated by VFDB (Table 3 and Appendix A). They belonged to nine virulence-related function categories, such as invasion, adherence, immune modulation, exotoxin, and so forth.

The three virulence genes of the UGLMO were *prfA, actA*, and *aut*. The pleiotropic regulatory factor (PrfA) is known as the principal regulator of LIPI-1, which expresses PlcA, PlcB, ActA, and Mpl [47]. The assembly-inducing protein (ActA) is greatly induced in the host cytoplasm and activates the ARP2/3 complex to polymerize actin and spread the bacteria from one cell to another [48]. The surface protein Auto with autolytic activity encoded by *aut* is required for LMO’s ability to enter into eukaryotic cells and necessary for its full virulence in vivo [49]. For the UGLIV, four virulence factor genes were identified, including the immune modulation genes internalin C (InlC), hexose phosphate transport protein (Hpt), sphingomyelinase-c (SmcL), and cytolysin regulator R2 (CylR2). The InlC protein can dampen the innate immune response and perturb the apical cell junctions, which promotes the cell-to-cell spread of *Listeria* [50,51]. The Hpt helps intracellular pathogens to exploit hexose phosphates from the host cell as a source of energy in order to promote bacteria growth within the cell. Its absence leads to the impaired intracellular growth of *listeria* and its reduced virulence in mice [52]. The SmcL mediates the membrane disruption of the phagosomes and the release of LIV into the cytosol [53].

It was noted that numerous adherence-, invasion-, and immune-modulation-related genes were shared by LMO and LIV, including *inlA, inlB, IlpA, ami, inlF, tufA, cps4I, gndA,* and *hasC*. They are critical genes that mediate *Listeria* pathogens’ adherence to and entry into the host cells. For instance, the N-terminal 15 leucine-rich repeat (LRR) region of InlA interacts with the E-cadherin of the host cell surface protein, which promotes LMO entry into the host cell [54]. Autolysin amidase, the product of *ami*, interacts with the glycosaminoglycans of the host cell to promote efficient adhesion to, and colonization in, the mouse hepatocytes of LMO [55,56]. The InlF, which can bind to surface-localized vimentin, is essential for LMO’s invasion of the host cell in brain infections [57]. Furthermore, four nutritional/metabolic factor genes (*mgtB, hbp1/svpA, hbp2, isdE*) and three exotoxin genes (*plcA, hly, plcB*) were detected in UGLP. It has been found that Hbp1 and Hbp2 are responsible for the acquisition of haem iron [58]. Specifically, Hbp1 can promote LMO’s escape from the phagosomes of macrophages [59]. The exotoxin factor phospholipase C (PlcA) and lecithinase (PlcB) cooperate with listeriolysin O (Hly) to mediate the dissolution of the membranes of phagosomes and the bacteria’s efficient escape to the cytoplasm [60,61]. Finally, exotoxin Mpl and the post-translational modification proteins GtcA and PrsA2 were found in the UGLP. Zinc metalloprotease (Mpl) is necessary for the maturation of PlcB [62]. The PrsA2 is a post-translocation chaperone, which mediates the secretion and maturation of the virulence proteins of *listeria* [63]. The glycosylation protein GtcA is involved in the modification of cell wall teichoic acid together with galactose and glucose and is related to cellular invasiveness [64].

#### 3.1.7. Ethanolamine-Metabolic-Related Proteins Enriched in the Protein–Protein Interaction Network of *Listeria* Pathogens’ Unique Genes

The protein–protein interaction network plays an important role in accomplishing various kinds of vital movements in living organisms. To further assess the associations between the unique genes of the *Listeria* pathogens, we performed a PPI analysis of the UGLP, UGLMO, and UGLIV gene sets using STRING, respectively. There were 189 genes included in the UGLMO PPI network (Figure 5A, Appendix A). They were clustered into a cluster, which indicated that they had strong associations in terms of their physical interactions or functional associations. Seventy-seven (77/189, 40.74%) of them were related to basic substance transport and metabolism, including six ABC-type transport system proteins (orf_2359, orf_2360, orf_2361, orf_135, orf_862, orf_849) and fourteen PTS system proteins (orf_1978, orf_1979, orf_21, orf_23, orf_2662, orf_2663, orf_2664, orf_2814, orf_2816, orf_297, orf_396, orf_398, orf_426, orf_632), as well as numerous enzymes related to energy metabolism. Moreover, thirty-six of them were related to genetic information processing, including nine DNA-binding transcriptional regulators. These regulators control gene expression in a variety of metabolic processes in bacteria, enabling them to adapt to complex and changeable environments. For example, the deletion or over-expression of the multiple antibiotic resistance regulator (MarR) of *Chromobacterium violaceum* significantly changed the gene expression of the oxidative stress response (*ohrA*/*ohrR*), anaerobic nitrate respiration (*narK1K2* and *narGHJI*), and biofilm formation (*RTX*) [65]. Furthermore, twenty-one proteins in the network were related to cell wall/membrane/envelope biogenesis and cell motility. Three flagella-related proteins (orf_708, orf_1077, orf_2706) and two cell-wall-biosynthesis-related proteins (orf_933, orf_2562) were clustered together in the network. It has been proved that flagellar motility is critical for *L. monocytogenes* biofilm formation and host cell invasion [66,67].

For the UGLVI set, there were 152 proteins shown in the PPI network, with one core network and four small networks (Figure 5B, Appendix A). Similar to the UGLMO, most of the proteins in the UGLVI PPI network (68/152, 44.73%) were related to basic substance transport and metabolism, enriched with ABC-type transport systems, PTS transporters, and energy metabolism enzymes. There were 22 (13.16%) proteins related to genetic information processing, including eight DNA-binding transcriptional regulators and six InlC proteins (orf_2796, orf_2507, orf_1673, orf_1678, orf_1682, orf_347), which represent an important factor in the promotion of intercellular invasion [51]. In addition, there were 22 proteins associated with the bacteria–environment interaction, including antibiotic resistance proteins (orf_1153, orf_1180, orf_1480), sensor proteins (orf_1132, orf_1136, orf_1898), and so forth.

For the UGLP gene set, we performed a functional enrichment analysis using the STRING tools. It is interesting to note that the ethanolamine-metabolic-process-related proteins were markedly enriched in the network (60/81, *P_adj_* < 0.001) (Appendix A, Appendix A). Ethanolamine is a compound of mammalian cell membranes, and some pathogens can be used as a source of carbon and/or nitrogen. Zeng et. al [68] found that ethanolamine utilization, which is driven by bacterial microcompartments, is essential for the anaerobic growth of LMO. The central genes of the ethanolamine metabolic process are *eutB* and *eutC* (orf_1178 and orf_1179, referenced to *L. monocytogenes* EGD-e 169963; orf_1755 and orf_1754, referenced to *L.ivanovii* 1638), which encode the ethanolamine ammonia-lyase [69]. In addition, five (referenced to *L. monocytogenes* EGD-e 169963) and seven (referenced to *L.ivanovii* 1638) accessory genes related to *eutB* and *eutC* were found in the PPI network [69]. It was found in *L. monocytogenes* that all the *eut* genes were strongly upregulated in the intestines of infected mice, and the *eut* operon was upregulated during intracellular growth in the cytosol of human colon epithelial cells, but the *eutB*-deleted strains proved to be defective in terms of intracellular growth [70,71]. The upregulation of the *eut* operon was also observed in *L. monocytogenes* that were co-cultured with cheese rind bacteria or grown on vacuum-packed, cold-smoked salmon, which suggests a possible role of the ethanolamine metabolism in the adaption to stress conditions [72,73]. However, more direct evidence is needed to make clear whether the ability to metabolize ethanolamine contributes to its pathogenicity through a specific mechanism.

As analyzed above, the majority of the unique proteins of *Listeria* pathogens have direct or indirect connections with their pathogenic lifestyle. These unique genes of LMO and LIV are responsible for their differences in niche adaption and pathogenicity. However, the 33 proteins shown in Figure 5A, 49 proteins in Figure 5B, and 139 proteins in Appendix A have no explicit function information, which should be studied in the future.

#### 3.1.8. The Distribution of Antibiotics Resistance Genes in the *Listeria* Species

With the use of the generally applied antibiotics, the antibiotic resistance of food-borne strains in many countries has increased [74]. The CARD database was used to investigate the antibiotic resistance gene distribution of the *Listeria* species. As shown in Figure 6, a total of 21 genes exist in the 307 *Listeria* genomes, which belong to 13 drug classes with five resistance mechanisms. The glycopeptide antibiotic resistance genes (*vanTG*, *vanYB*), peptide antibiotic resistance genes (*Lmon_mprF*), phosphonic acid antibiotic resistance genes (*FosX*), and lincosamide antibiotic gene *lin* were the most widely attested antibiotic resistance genes in *Listeria*. LMO, *L. welshimeri,* and *L. innocua* had similar types of antibiotic genes. Importantly, *L. innocua* had the most abundant antibiotic resistance genes, and as many as eight of them were found to exist in the plasmid. These results follow the survey data on antibiotic resistance among *Listeria* [75,76]. The high-level presence of resistance genes in other *Listeria* species, especially *L. innocua*, could increase the possibility of the acquisition of resistance by LMO through horizontal gene transfer in the future.

#### 3.1.9. The Occurrence of Different Types of CRISPR-Cas Systems in *Listeria*

The CRISPR-Cas systems are bacterial-adaptive antivirus immunity systems protecting bacteria from the infection of bacteriophages, which are also associated with the virulence and pathogenicity of pathogens [77,78]. In this study, we investigated the CRISPR-Cas systems of 307 *Listeria* genomes. Five types of CRISPR-Cas systems were found, including I-B, II-A, II-C, III-B, and VI-A, which could be divided into seven sub-types (Table 4). About one-third of the LMO genomes (90/275) had at least one CRISPR-Cas system. Among them, five sub-type CRISPR-Cas systems, except for II-C_n1 and VI-A_n2, were found in it, and I-B_n1 (36.89%) II-A_n1 (49.51%) were the most prevalent subtypes. The subtypes I-B_n2 and III-B_n1 were only detected in the LMO. The subtype II-C_n1 was only detected in LIV, and VI-A_n2 was only detected in *L. seeligeri*. Because of the small numbers of genomes of *L. grayi* and *L. welshimeri*, no CRISPR-Cas system was found in them.

#### 3.1.10. The Characterization of Plasmids from the *Listeria* Species

The plasmid was previously detected in many *Listeria* wildtype strains and is responsible for the transferable antibiotic resistance of LMO [79,80]. A total of 60 plasmids were detected in the 307 *Listeria* genomes, which were classified into 11 types based on the plasmidfinder results (Table 5 and Appendix A). The lengths of the plasmids ranged from 2776 bp (CP045971.1) to 152,337 bp (CP022021.1). Nine types of plasmid were found in the LMO, and half of them (26/51) were PLM33. We also investigated the antibiotic resistance genes of the plasmids. The results showed that the plasmids of 15 LMO carried qacG/qacJ genes, which are related to the resistance of disinfecting agents and antiseptics through the antibiotic efflux mechanism. It is interesting to note that the plasmids of *L. innocua* carried as many as eight resistance genes, including APH(3’)-IIIa, ANT(6)-Ia, ErmB, tet(S), dfrG, catA8, lnuA, and lsaE, which suggested that there was a risk of drug resistance transmission among the *Listeria* pathogens through horizontal gene transfer. In addition, we did not find any other *Listeria* plasmids carrying antibiotic resistance genes.

### 3.2. Formatting of Mathematical Components

Core genome size profile model:(1)$$y=A_{1}e^{B1x}+C_{1}$$
where y is the core genome size; x is the number of the genome; and *A*_1_, *B*_1_, and *C*_1_ are free parameters.

Pan-genome size profile model:(2)$$y=A_{2}x^{B2}+C_{2}$$
where y is the pan-genome size; x is the number of the genome; and *A*_2_, *B*_2_, and *C*_2_ are free parameters.

## 4. Conclusions

In this study, we presented a genus-wide comparative genomic study of *Listeria* species to investigate the genetic basis of the phenotypes affecting the pathogenicity and niche adaption of *Listeria* pathogens. The core and pan-genome analysis was performed based on the genomes of LMO, LIV, and NPLS. The core and pan-genome sizes of all the *Listeria* genomes, LMO, LIV, and NPLS were 975/7701, 2060/5382, 2118/3194, and 1203/5308, respectively. The regression of the pan-genome size using the Heaps’ law model showed that LMO had an open genome, which can be understood as the genetic basis for its adaption to diverse environments. In addition, the numbers of UGLMO, UGLIV, and UGLP were 204, 242, and 756, respectively. The function annotation results indicated that numerous carbohydrate-metabolism-related proteins for various carbon sources and virulence-associated proteins were included in the unique gene sets, which play important roles in their pathogenicity and niche adaption. Furthermore, the PPI network analysis of UGLMO, UGLIV, and UGLP showed that the majority of these genes were closely connected. Numerous ABC-type transport systems, PTS transporters, and energy metabolism enzymes were clustered in the UGLMO and UGLIV PPI networks, as well as DNA-binding transcriptional regulators, flagella-related proteins, and so on. Interestingly, ethanolamine-metabolic-process-related proteins were significantly enriched in the UGLP network, which might be a significant factor in the pathogenicity of *Listeria* pathogens. However, 33 proteins of UGLMO, 49 proteins of UGLIV, and 139 proteins of UGLP in the PPI network still have no explicit function information, and these proteins might interact with others and should be investigated in the future.

## Figures and Tables

**Figure 1 pathogens-11-01430-f001:**
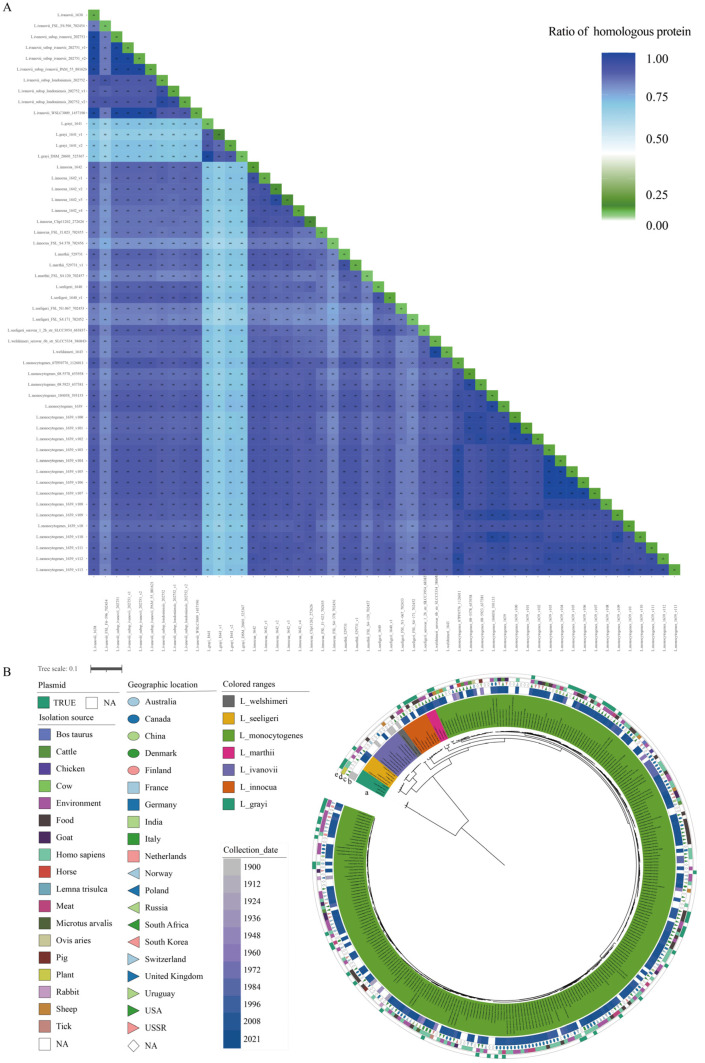
Homologous proteome analysis and phylogeny of the *Listeria* genus strains. (**A**) The percentages of homologous proteome among the *Listeria* strain pairs and within strains. The color depth of each block represents the percentage of the homologous proteome of the corresponding strain pairs. Twenty LMO genomes were selected as representatives in the heatmap. The number and percentage of the homologs between/within strains are presented in corresponding blocks. (**B**) Phylogenetic analysis of the *Listeria* strains examined in the present study. (a) The phylogenetic tree is based on the concatenated sequences of 573 single-copy core genes. The different species are shown in different colors. (b) The collection dates of the strains are shown in gradient colors. (c) The geographic locations of the strains isolated are shown with combinations of different shapes and colors. (d) The isolation sources of strain are shown in different colors. (e) The presence of plasmids in each strain is shown with a green color bar.

**Figure 2 pathogens-11-01430-f002:**
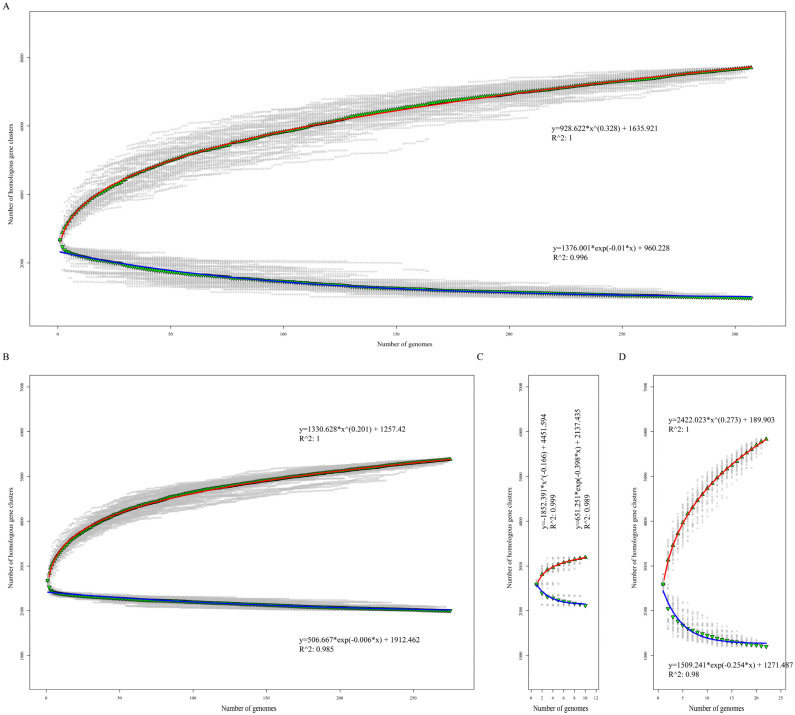
Core and pan-genome size evolution. Accumulation curves of the core and pan-genomes of 307 *Listeria* strains (**A**), 275 *L. monoeytogenes* (**B**), 10 *L. ivanovii* (**C**), and 22 non-pathogenic *Listeria* strains (**D**), respectively. Blue and red curves represent the core and pan-genome fitting curves of each group. The ▼ and ▲ represent the average core and pan-genome size of each group with 100 random permutations.

**Figure 3 pathogens-11-01430-f003:**
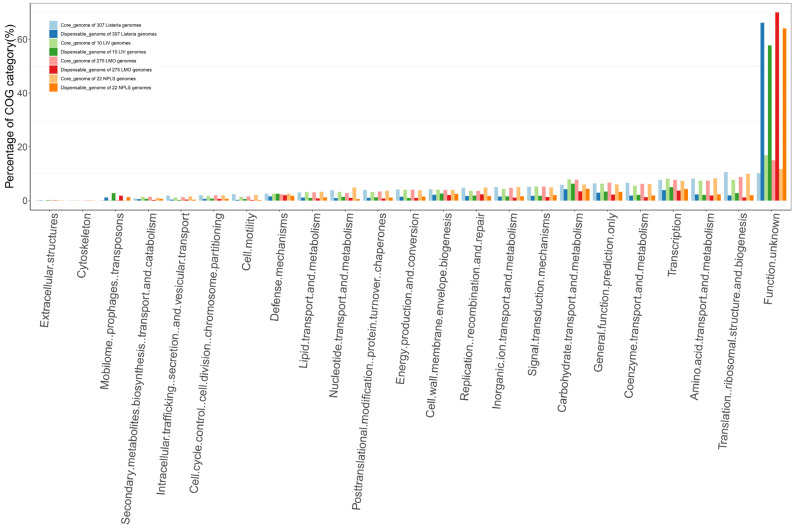
Function categories of the core and accessory genomes of each *Listeria* strain group in the COG and EggNOG databases. The COG categories of the core and accessory genomes of 307 *Listeria* genomes, 10 LIV genomes, 275 LMO genomes, and 22 NPLS genomes are shown in the graph with paired colors, respectively.

**Figure 4 pathogens-11-01430-f004:**
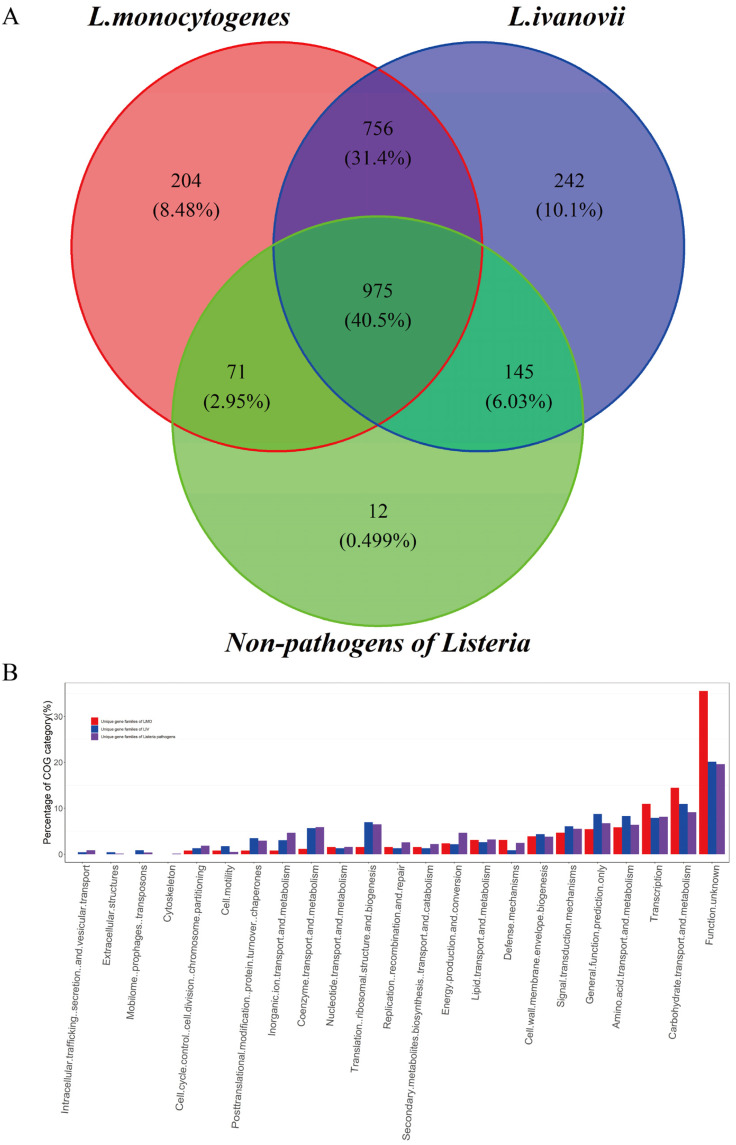
Function categories of the unique gene families of LMO, LIV, and both. (**A**) The Venn diagram for the gene sets of *L. monocytogenes*, *L. ivanovii*, and non-pathogens of *Listeria*. The number and proportion of the gene families are shown in the graph. (**B**) Function categories of the unique gene sets of LMO, LIV, and both according to the COG and EggNOG databases. The COG categories of 204 LMO unique genes, 242 LIV unique genes, and 756 genes that were shared by LMO and LIV but absent in NPLS are shown in the graph with different colors, respectively.

**Figure 5 pathogens-11-01430-f005:**
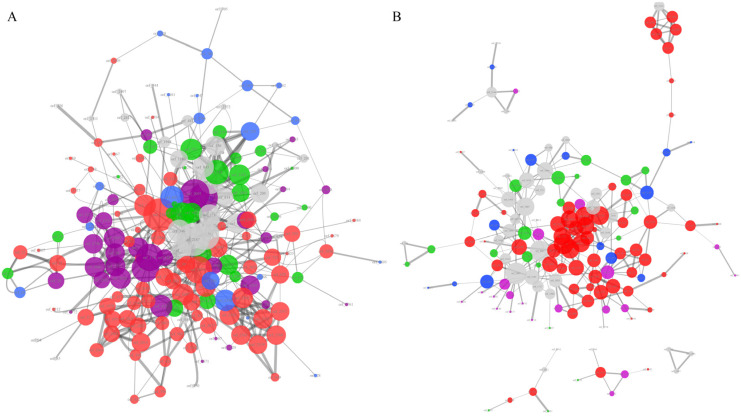
The protein–protein interaction network analysis of unique gene families of the LMO and LVI genomes. The PPI network of LMO (**A**) referenced to *L. monocytogenes* EGD-e 169963 and the names of the nodes corresponding to Appendix A. The PPI network of LMO (**B**) referenced to *L. ivanovii* 1638 and the names of the nodes corresponding to Appendix A. Only those networks with a number of nodes greater than 3 are shown in the graph. The edges of the network represent the relationships of protein–protein interaction, and their line thickness represents the confidence of the relationships between nodes. The size of the nodes represents the degree of the nodes. Different colors of the nodes reflect the different protein function categories: the red represents basic substance transport and metabolism; the purple represents genetic information processing, including replication, transcription, and translation; the blue represents the cellular processes, including cell wall/membrane/envelope biogenesis and cell motility; the green represents the bacteria–environment interaction, including signal transduction, extracellular structures, and defense mechanism; and the grey represents the proteins without meaningful function annotations.

**Figure 6 pathogens-11-01430-f006:**
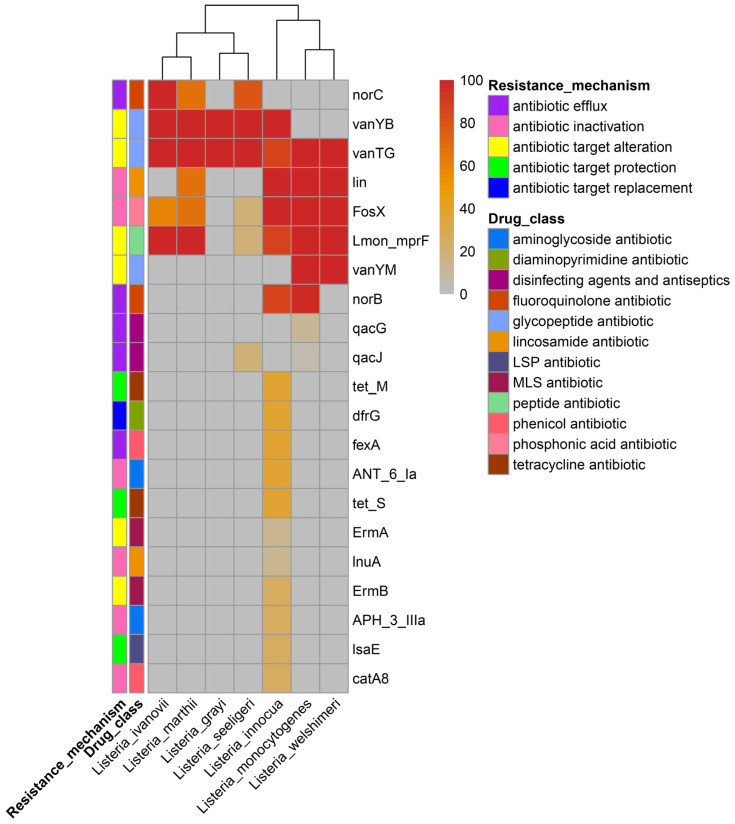
The distribution of antibiotics resistance genes in the *Listeria* species. The percentages of genomes that contain antibiotic resistance genes for each species are shown in the heatmap. The resistance mechanisms of the genes and corresponding drug class are shown in the graph. The LSP antibiotic drug class refers to lincosamide antibiotic, streptogramin antibiotic, and pleuromutilin antibiotic; the MLS refers to macrolide antibiotic, lincosamide antibiotic, and streptogramin antibiotic.

**Table 1 pathogens-11-01430-t001:** Characteristics of *Listeria* species used in the study.

Species	Number of Genomes	Genome Size (bp) ^1^	GC Content (%) ^1^	Number of Scaffolds	Number of Plasmids	CDS ^1,2^	rRNA ^1^	tRNA ^1^	Isolation Source
*L.grayi*	4	2,600,550 (2,598,321, 2,751,104)	41.52 (41.47, 41.73)	1–4	0–1	2609 (2561, 2982)	12 (7, 18)	58 (52, 65)	Homo sapiens, Plant
*L. innocua*	8	3,019,705 (2,889,727, 3,115,943)	37.38 (34.65, 37.48)	1	0–1	2934 (2622, 3130)	18 (0, 18)	67 (24, 67)	Food, Meat, Environment
*L. ivanovii*	10	2,924,216 (2,893,311, 3,133,385)	37.11 (34.59, 37.15)	1	0	2808 (2682, 2924)	18 (1, 18)	67 (27, 67)	Food, Sheep
*L. marthii*	3	2,884,551 (2,840,389, 2,947,729)	38.62 (36.79, 38.74)	1	0	2759 (2597, 2863)	18 (2, 18)	67 (28, 67)	Environment
*L. monocytogenes*	275	2,964,472 (2,776,517, 3,243,301)	37.98 (37.74, 38.29)	1	0–1	2906 (2730, 3323)	18 (4, 20)	67 (49, 72)	Cow, Chicken, Food, Homo sapiens, Pig, Goat, Meat, Horse, Rabbit, Environment
*L. seeligeri*	5	2,927,941 (2,797,636, 3,140,155)	37.30 (35.17, 37.38)	1	0–1	2779 (2590, 2846)	8 (1, 18)	63 (24, 67)	Environment
*L. welshimeri*	2	2,814,134 (2,814,130, 2,814,137)	36.35 (36.35, 36.35)	1	0	2,780 (2778, 2781)	18 (18, 18)	66 (66, 66)	Food
Total	295	2,959,738 (2,598,321, 3,243,301)	37.97 (34.59, 41.73)	1–4	0–1	2902 (2561, 3323)	18 (0, 20)	67 (24, 72)	-

Note: ^1^ Median (Range). ^2^ CDS, coding sequence.

**Table 2 pathogens-11-01430-t002:** Unique transporter genes of *Listeria* pathogens.

Type of Transporter	Type of Substrate	Protein Name	ORFs ^1^ (Referenced to *L. monocytogenes* EGD-e_169963)	ORFs (Referenced to *L. ivanovii* 1638)
Phosphotransferase system	cellobiose	CelC, CelA, CelB	orf_1724, orf_2698, orf_915, orf_2723, orf_916	orf_1158, orf_242, orf_2009, orf_186, orf_2008
	fructose	FrwB	orf_2144, orf_397, orf_425, orf_633	orf_2291, orf_722
		FrwC	orf_398, orf_426, orf_632	NE ^2^
	mannose/fructose/N-acetylgalactosamine	AgaB, ManY, ManZ, ManX	orf_2009, orf_22, orf_2008, orf_783, orf_96, orf_2007, orf_24, orf_782, orf_2004, orf_785	orf_2656, orf_869, orf_2151, orf_2780, orf_870, orf_2152, orf_871, orf_2149, orf_874
	sorbitol	SrlB, SrlE, SrlA,	orf_542, orf_543, orf_544	orf_2404, orf_2403, orf_2402
	galactitol	SgaB	orf_1979, orf_2663	NE
	mannitol	MtlA	orf_2816	NE
ABC-type transport system	sugar transport	MalK, UgpA	orf_2132, orf_2856, orf_861	orf_2613, orf_2068, orf_735, orf_75
	polysaccharide	LplB	orf_2016	orf_862
	glycerol-3-phosphate	UgpB	orf_2014, orf_2857, orf_860	orf_864
		UgpE	orf_2015, orf_2855	NE
	multidrug transport	MdlB, CcmA, YadH	orf_106, orf_1135, orf_2766, orf_607, orf_608, orf_2235, orf_980, orf_987, orf_981, orf_988	orf_143, orf_1807, orf_2332, orf_2333, orf_2767, orf_1934, orf_2207, orf_635, orf_1932, orf_1933, orf_2206
	Mn^2+^/Zn^2+^/Fe^3+^/Co^2+^/Mo^2+^	PotA, CbiM, CbiN, FepB, FepD, NlpA, ZnuC, ZnuB, ModA, ModB, ZnuA, FepC	orf_1040,orf_1207, orf_1208, orf_2192, orf_1964, orf_281, orf_1453, orf_1855, orf_1452, orf_1854, orf_1042, orf_1041, orf_1676, orf_2190	orf_1727, orf_1726, orf_678, orf_915, orf_2606, orf_1028, orf_1450, orf_1029, orf_1451,orf_1878, orf_1879, orf_1206, orf_680
	antimicrobial peptide, polar amino acid, methionine, lipoprotein,	SalY, GlnQ, AbcC, LolD, MetP, LolC	orf1_192, orf1_2259, orf1_2358, orf1_1511, orf1_279, orf1_1227	orf1_2529, orf1_2703, orf1_611, orf1_2607, orf1_1393, orf1_2608, orf1_1700
	Others	EcfA2, DdpA, Uup, CcmA, CydD	orf1_133, orf1_1210, orf1_2731, orf1_920, orf1_745, orf1_1134	orf1_2004, orf1_2751, orf1_1724, orf1_2188, orf1_178, orf1_1808

Note: ^1^ ORFs, open reading frames. ^2^ NE, not existing.

**Table 3 pathogens-11-01430-t003:** Unique virulence genes of *Listeria* pathogens.

Function Category	Gene Name	UGLMO ^1^	UFLIV ^2^	UGLP(LMO) ^1^	UGLP(LIV) ^2^	Gene’s Description of VFDB
Regulation	*prfA*	orf1_197				Listeriolysin-positive regulatory protein
Motility	*actA*	orf1_201				Actin-assembly-inducing protein precursor
Invasion	*aut*	orf_1077				Autolysin
	*inlA*			orf_431	orf_2509	Internalin A
	*inlB*			orf_432	orf_2506	Internalin B
	*inlC*		orf_348			Internalin C
Immune modulation	*cps4I*			orf_2549	orf_401	Capsular polysaccharide biosynthesis protein Cps4I
	*gndA*			orf_1382	orf_1532	NADP-dependent phosphogluconate dehydrogenase
	*hasC*			orf_1079	orf_1839	UTP–glucose-1-phosphate uridylyltransferase
Nutritional/metabolic factor	*mgtB*			orf_2703	orf_239	Mg^2+^ transport protein
	*hbp1/svpA*			orf_2194	orf_676	Haemoglobin-binding protein 1
	*hbp2*			orf_2193	orf_677	Hypothetical protein
	*isdE*			orf_2192	orf_678	Iron-regulated surface-determinant protein E
	*hpt*		orf1_2093			Hexose phosphate transport protein
Exoenzyme	*smcL*		orf1_1675			Sphingomyelinase-c
	*mpl*			orf_200	orf_2695	Zinc metalloproteinase precursor
Exotoxin	*cylR2*		orf1_2557			Cytolysin regulator R2
	*plcB*			orf_202	orf_2693	Phospholipase C
	*hly*			orf_199	orf_2696	Listeriolysin O precursor
	*plcA*			orf_198	orf_2697	Phosphatidylinositol-specific phospholipase c
Adherence	*tufA*			orf_2666	orf_278	Elongation factor Tu
	*ami*			orf_2570	orf_1680	Autolysin amidase, adhesin
	*inlF*			orf_407	orf_474	Internalin F
Post-translational modification	*gtcA*			orf_2561	orf_385	Wall teichoic acid glycosylation protein
	*prsA2*			orf_2227	orf_643	Post-translocation chaperone

Note: ^1^ Referenced to *L. monocytogenes* EGD-e_169963; ^2^ referenced to *L. ivanovii* 1638.

**Table 4 pathogens-11-01430-t004:** The identified types of CRISPR-Cas system in the 307 *Listeria* genomes.

Species	Number of Each Cas-CRISPR System Type							
	I-B_n1	I-B_n2	II-A_n1	II-A_n2	II-C_n1	III-B_n1	VI-A_n2	Total
*L. grayi*	0	0	0	0	0	0	0	0
*L. innocua*	0	0	4	0	0	0	0	4
*L. ivanovii*	2	0	0	2	2	0	0	6
*L. marthii*	1	0	0	0	0	0	0	1
*L. monocytogenes*	38	3	51	10	0	1	0	103
*L. seeligeri*	3	0	1	0	0	0	1	5
*L. welshimeri*	0	0	0	0	0	0	0	0
Total	44	3	56	12	2	1	1	119

**Table 5 pathogens-11-01430-t005:** The plasmid distribution in the 307 *Listeria* genomes.

Species	Number of Each Plasmid Type											
	LMIVRS16815	M640p00130	M643p00680	pLGUG1	pLI100	pLIS3	pLIS5	pLM33	pLM5578	pLMIV	pLMUKDL7	Total
*L. grayi*	0	0	0	3	0	0	0	0	0	0	0	3
*L. innocua*	0	2	0	0	1	0	0	2	0	0	0	5
*L. monocytogenes*	1	6	3	4	0	2	0	26	6	2	1	51
*L. seeligeri*	0	0	0	0	0	0	1	0	0	0	0	1
Total	1	8	3	7	1	2	1	28	6	2	1	60

## Data Availability

The data used to support the findings of this study are available from the corresponding author upon request.

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
