# Peer review of "Comparative Genomics Reveal the Utilization Ability of Variable Carbohydrates as Key Genetic Features of Listeria Pathogens in Their Pathogenic Lifestyles"

_pathogens, 2022, doi:10.3390/pathogens11121430_

Round 1

Reviewer 1 Report

Dear authors,

The original manuscript entitled “Comparative genomics of Listeria pathogens and non-pathogens reveals the unique genetic composition related to their pathogenic lifestyles” is suitably developed, structured and written by Lu et al., in appropriate English with a clear structure. This manuscript is unique, novel and valuable. They conducted a bioinformatics big data analysis study and comparative genomic analysis based on 275 L. monocytogenes, 10 L. ivanovii, and 22 non-pathogenic Listeria strains. The obtained results are very interesting. To improve this valuable manuscript, it is strongly recommended to

-        Add antibiotic resistance gene detection and discuss the results

-        Investigate the virulence factor encoding genes in a separate section and present the data in a separate table with a suitable discussion

-        Discuss the pathogenesis and molecular mechanisms of Listeria pathogenicity regarding the virulence factors of this pathogens

-        Detect the plasmids using the plasmid databank and discuss the results

-        Detect the CRISPR arrays and related genes using an appropriate online platform and discuss the results

Please address all of these revisions. After revising, this manuscript must be reconsidered. 

Author Response

Response to Reviewer 1 Comments:

Dear authors,

The original manuscript entitled “Comparative genomics of Listeria pathogens and non-pathogens reveals the unique genetic composition related to their pathogenic lifestyles” is suitably developed, structured and written by Lu et al., in appropriate English with a clear structure. This manuscript is unique, novel and valuable. They conducted a bioinformatics big data analysis study and comparative genomic analysis based on 275 L. monocytogenes, 10 L. ivanovii, and 22 non-pathogenic Listeria strains. The obtained results are very interesting. To improve this valuable manuscript, it is strongly recommended to:

Concern #1: Add antibiotic resistance gene detection and discuss the results

Response: As suggested, the Resistance Gene Identifier (RGI-v6.0.1) was used for detecting the antibiotic resistance genes of Listeria genomes based on the Comprehensive Antibiotic Resistance Database (CARD version 3.2.5). The results were summarized and shown with a heatmap (Figure 6). As detailed in Figure 6 in the revised manuscript, a total of 21 genes existed in the 307 Listeria genomes, which belong to 13 drug classes. The glycopeptide antibiotic resistance genes (vanTG, vanYB), peptide antibiotic resistance genes (Lmon_mprF), phosphonic acid antibiotic resistance genes (FosX), and lincosamide antibiotic gene lin were the most widely existing antibiotic resistance genes in Listeria. LMO, L. welshimeri, and L. innocua had similar types of antibiotic genes. Importantly, L. innocua had the most abundant antibiotic-resistance genes and as many as eight of them were found to exist in the plasmid. These results were following the survey data on antibiotic resistance among Listeria [51,52]. The high-level presence of resistance genes in other Listeria species, especially L. innocua, increases the possibility of the acquisition of resistance by LMO according to the horizontal gene transfer in the future. Those information has been added to the revised manuscript (section 3.1.8, lines 465-486).

Figure 6. The distribution of antibiotics resistance genes of Listeria species. The percentages of genomes that contain antibiotic resistance genes of each species were shown in the heatmap. The resistance mechanism of genes and corresponding drug class were shown in the graph. The LSP antibiotic of drug class means lincosamide antibiotic, streptogramin antibiotic, and pleuromutilin antibiotic; the MLS means macrolide antibiotic, lincosamide antibiotic, and streptogramin antibiotic.

Concern #2: Investigate the virulence factor encoding genes in a separate section and present the data in a separate table with a suitable discussion

Response: As suggested, we have investigated the virulence factor encoding genes of the unique gene dataset of Listeria pathogens using the VFDB database. And the results were summarized and listed in Table 3. There were 3 ORFs of UGLMO, 4 ORFs of UGLIV, and 18 ORFs of UGLP that could be annotated by VFDB. They belonged to nine virulence-related function categories, such as invasion, adherence, immune modulation, exotoxin and so forth. The summarized table and discussion of virulence factors have been added in the revised manuscript (section 3.1.6, lines 343-388).

Table 3. Unique virulence genes of Listeria pathogens

Function category

Gene name

UGLMO1

UFLIV2

UGLP(LMO)1

UGLP(LIV)2

Description in VFDB

Regulation

prfA

orf1_197

listeriolysin positive regulatory protein

Motility

actA

orf1_201

actin-assembly inducing protein precursor

Invasion

aut

orf_1077

autolysin

inlA

orf_431

orf_2509

Internalin A

inlB

orf_432

orf_2506

Internalin B

inlC

orf_348

internalin C

Immune modulatio

cps4I

orf_2549

orf_401

capsular polysaccharide biosynthesis protein Cps4I

gndA

orf_1382

orf_1532

NADP-dependent phosphogluconate dehydrogenase

hasC

orf_1079

orf_1839

UTP--glucose-1-phosphate uridylyltransferase

Nutritional/Metabolic factor

mgtB

orf_2703

orf_239

Mg2+ transport protein

hbp1/svpA

orf_2194

orf_676

Haemoglobin binding protein 1

hbp2

orf_2193

orf_677

hypothetical protein

isdE

orf_2192

orf_678

iron-regulated surface determinant protein E

hpt

orf1_2093

hexose phosphate transport protein

Exoenzyme

smcL

orf1_1675

sphingomyelinase-c

mpl

orf_200

orf_2695

Zinc metalloproteinase precursor

Exotoxin

cylR2

orf1_2557

cytolysin regulator R2

plcB

orf_202

orf_2693

phospholipase C

hly

orf_199

orf_2696

listeriolysin O precursor

plcA

orf_198

orf_2697

phosphatidylinositol-specific phospholipase c

Adherence

tufA

orf_2666

orf_278

elongation factor Tu

ami

orf_2570

orf_1680

autolysin amidase, adhesin

inlF

orf_407

orf_474

internalin F

Post-translational modification

gtcA

orf_2561

orf_385

wall teichoic acid glycosylation protein

prsA2

orf_2227

orf_643

post translocation chaperone

Note: 1 Referenced with L. monocytogenes EGD-e_169963;

         2 Referenced with L. ivanovii 1638.

Concern #3: Discuss the pathogenesis and molecular mechanisms of Listeria pathogenicity regarding the virulence factors of this pathogens

Response: As suggested, we have discussed the pathogenesis and molecular mechanisms of Listeria pathogenicity based on the virulence factor genes investigation results of unique genes of Listeria pathogens in section 3.1.6 (lines350-385). The discussion was as follow:

Three virulence genes of the UGLMO were prfA, actA and aut. The pleiotropic reg-ulatory factor (PrfA) is known as the principal regulator of LIPI-1, which expresses PlcA, PlcB, ActA and Mpl [47]. The assembly-inducing protein (ActA) is greatly induced in the host cytoplasm and activates the ARP2/3 complex to polymerize actin and spread the bacteria from one cell to another [48]. The surface protein Auto with autolytic activity encoded by aut is required for LMO entering into eukaryotic cells and necessary for its full virulence in vivo [49]. For the UGLIV, four virulence factor genes were identified, including immune modulation genes internalin C (InlC), hexose phosphate transport protein (Hpt), sphingomyelinase-c (SmcL) and cytolysin regulator R2 (CylR2). The InlC protein could dampen the innate immune response and perturb apical cell junctions, which promotes the cell-to-cell spread of Listeria [50,51]. The Hpt helps intracellular pathogens to exploit hexose phosphates from the host cell as a source of energy to promote bacteria growth within the cell. Absent of it lead to im-paired intracellular growth of listeria and reduced virulence in mice [52]. The SmcL mediates the membrane disruption of the phagosomes and the release of LIV into the cytosol [53].

 It was noted that plenty of adherence and, invasion and immune modulation-related genes were shared by LMO and LIV, including inlA, inlB, IlpA, ami, inlF, tufA, cps4I, gndA and hasC. They were critical genes that mediate Listeria pathogens' adherence and entry into the host cells. For instance, the N-terminal 15 leucine-rich repeat (LRR) region of InlA interacts with the E-cadherin of host cell surface protein, which promotes LMO entry into the host cell [54]. Autolysin amidase, the product of ami, interacts with the glycosaminoglycans of the host cell to promote efficient adhesion and colonization to mouse hepatocytes of LMO [55,56]. The InlF, which can bind to surface-localized vimentin, is essential for LMO invasion of the host cell in brain infection [57]. Furthermore, four nutritional/metabolic factor genes (mgtB, hbp1/svpA, hbp2, isdE) and three exotoxin genes (plcA, hly, plcB) were detected in UGLP. It has been found that Hbp1 and Hbp2 are responsible for the acquisition of haem iron [58]. Specifically, Hbp1 can promote LMO escape from the phagosomes of macrophages [59]. The exotoxin factor phospholipase C (PlcA) and lecithinase (PlcB) cooperate with listeriolysin O (Hly) to mediate the dissolution of the membrane of phagosomes and efficient escape to the cytoplasm [60,61]. Finally, exotoxin Mpl and post-translational modification proteins GtcA, PrsA2 were found in the UGLP. The zinc metalloprotease (Mpl) is necessary for the maturation of PlcB [62]. The PrsA2 is a post-translocation chaperone, which mediates the secretion and maturation of virulence proteins of listeria [63]. The glycosylation protein GtcA is involved in the modification of cell wall teichoic acid with galactose and glucose and is related to cellular invasiveness [64].

Concern #4: Detect the plasmids using the plasmid databank and discuss the results

Response: As suggested, plasmidfinder 2.1.6.1 was used to detect the plasmids of Listeria genomes and the results were validated with the plasmid database v2021_06_23_v2. The results were summarized and shown in Table 5. There was a total of 60 plasmids detected in the 307 Listeria genomes, which were classified into 11 types based on the plasmidfinder results (Table 5, S7). The length of plasmids ranged from 2776bp (CP045971.1) to 152337bp (CP022021.1). Nine types of plasmid were found in the LMO and half of them (26/51) were PLM33. We also investigated the antibiotic-resistance genes of plasmids. The results showed that the plasmids of 15 LMO carried qacG/qacJ genes, which related to the resistance of disinfecting agents and antiseptics with antibiotic efflux mechanism. It is interesting to notice that the plasmids of L. innocua carried as many as eight resistance genes, including APH(3')-IIIa, ANT(6)-Ia, ErmB, tet(S), dfrG, catA8, lnuA and lsaE, which suggested that there was a risk of drug resistance transmission of Listeria pathogens through the horizontal gene transfer. In addition, we did not find any other plasmids carrying antibiotic-resistance genes in these Listeria genomes. This information has been added to the revised manuscript (section 3.1.10, lines 504-520).

Table 5. The plasmids distribution in the 307 Listeria genomes

Species

Number of each plasmid type

LMIVRS16815

M640p00130

M643p00680

pLGUG1

pLI100

pLIS3

pLIS5

pLM33

pLM5578

pLMIV

pLMUKDL7

Total

L. grayi

0

0

0

3

0

0

0

0

0

0

0

3

L. innocua

0

2

0

0

1

0

0

2

0

0

0

5

L. monocytogenes

1

6

3

4

0

2

0

26

6

2

1

51

L. seeligeri

0

0

0

0

0

0

1

0

0

0

0

1

Total

1

8

3

7

1

2

1

28

6

2

1

60

Concern #5: Detect the CRISPR arrays and related genes using an appropriate online platform and discuss the results

Response: In response to this concern, we detected the CRISPR-Cas systems of 307 Listeria genomes using CRISPRCasFinder version 4.2.20. To get the functional CRISPR-Cas system, the presence of both the CRISPR sequence and Cas genes was considered as evidence for an actual CRISPR-Cas system and used for further analysis. As listed in Table 4, five types of CRISPR-Cas systems were found, including I-B, II-A, II-C, III-B, and VI-A, which could be divided into seven sub-types. About one-third of LMO genomes (90/275) had at least one CRISPR-Cas system. Among them, five sub-type CRISPR-Cas systems, except II-C_n1 and VI-A_n2, were found and I-B_n1 (36.89%), and II-A_n1 (49.51%) were the most prevalent subtypes. The subtypes I-B_n2 and III-B_n1 were only detected in the LMO. The subtype II-C_n1 was only detected in LIV and VI-A_n2 was only detected in L. seeligeri. Because of the few genomes of L. grayi and L. welshimeri, no CRISPR-Cas system was found in them. Those results were added in the revised manuscript (section 3.1.9, lines 487-503).

Table 4. The identified types of CRISPR-Cas system in the 307 Listeria genomes

Species

Number of each Cas-CRISPR system Type

I-B_n1

I-B_n2

II-A_n1

II-A_n2

II-C_n1

III-B_n1

VI-A_n2

Total

L. grayi

0

0

0

0

0

0

0

0

L. innocua

0

0

4

0

0

0

0

4

L. ivanovii

2

0

0

2

2

0

0

6

L. marthii

1

0

0

0

0

0

0

1

L. monocytogenes

38

3

51

10

0

1

0

103

L. seeligeri

3

0

1

0

0

0

1

5

L. welshimeri

0

0

0

0

0

0

0

0

Total

44

3

56

12

2

1

1

119

Please address all of these revisions. After revising, this manuscript must be reconsidered.

Reviewer 2 Report

Here I present the revision of the manuscript entitled "Comparative genomics of Listeria pathogens and non-pathogens reveals the unique genetic composition related to their pathogenic lifestyles". This is an interesting manuscript that provides deep insights into the genetic composition that are relevant. It joins high volume of results which are represented logically and understandably. Overall, I think that it is an adequate study to be published in the journal, although with some suggestions being applied. These are as follows: 

General suggestions:

1) I think that the title is quite general. With the number of results obtained, they can improve the title by adding more concrete information. 

2) I consider that the importance/relevance of the study should be included with more detail and should be reinforced through the manuscript. In the introduction (Line 69-71) is, more or less, covered: “The comparative genomic analysis of different species or strains with different phenotypes could help us get a more comprehensive and better understanding of the genetic basis underlying the unique phenotypes.”. However, more details can be given, as well as it can be reinforced on other parts of the manuscript. 

3) General objective and specific objectives of the study should be included in the last part of the introduction. 

Minor suggestions:

1) Authors indicated that there are a total of 29 species of Listeria, however, there are a total of 30 identified Listeria spp. In this case, L. ilorinensis y L. swaminathanii have been discovered in 2022. Maybe one of them was discovered and included after authors consulted LPSN. Authors should correct that (Line 41) as well as in Line 50 they have to correct the sum of non-pathogenic Listeria spp. (28 in total). 

2) Line 41: pH rather than PH.

3) Table 1: The definition of the acronyms should be included at the end of the table.

4) Table 1 (Line 93) – Listeria is written without italic spelling. Names of microorganisms should be in italic. Please change that. This is a repeated mistake that is found in different parts of the manuscript:

“Line 160, Line 161, Line 183 (L. seeligeri), Line 238, Line 267… among others”. Revise the whole manuscript to correct that. 

5) Line 147: Numbers that start after a point should be written with letters (i.e. 275)

6) Line 218: Should that not be Figure 2?

7) Line 343: “Deletion” with “d” in lowercase.

8) Bibliography requires a whole careful reading and correction. Names of the microorganisms are not in italic; some of the titles of the articles start with an uppercase in all the word while others everything is un lowercase. All of that requires correction.  

Author Response

Response to Reviewer 2 Comments:

Here I present the revision of the manuscript entitled "Comparative genomics of Listeria pathogens and non-pathogens reveals the unique genetic composition related to their pathogenic lifestyles". This is an interesting manuscript that provides deep insights into the genetic composition that are relevant. It joins a high volume of results which are represented logically and understandably. Overall, I think that it is an adequate study to be published in the journal, although with some suggestions being applied. These are as follows:

General suggestions:

Concern #1: I think that the title is quite general. With the number of results obtained, they can improve the title by adding more concrete information.

Response: In response to this concern, we have rewritten the title as “Comparative genomics reveals utilization ability of variable carbohydrates as key genetic features of Listeria pathogens for their pathogenic lifestyles” in the revised manuscript (lines 2-6).

Concern #2: I consider that the importance/relevance of the study should be included with more detail and should be reinforced through the manuscript. In the introduction (Line 69-71) is, more or less, covered: “The comparative genomic analysis of different species or strains with different phenotypes could help us get a more comprehensive and better understanding of the genetic basis underlying the unique phenotypes.”. However, more details can be given, as well as it can be reinforced on other parts of the manuscript.

Response: As suggested, we have rewritten the sentences with more details to better demonstrate the significance of the study. As shown in lines 73-79, the importance/relevance of the study has been expressed as “In recent years, along with the large-scale application of next-generation sequence and third-generation sequence technology, plenty of Listeria genomes have been sequenced and shared. The comparative genomic analysis of Listeria pathogens and non-pathogenic genomes would improve our understanding of the genetic mechanisms for them to adapt to diverse environments and their pathogenic lifestyles, which will further promote the diagnosis and treatment of listeria infections.”.

Concern #3: General objective and specific objectives of the study should be included in the last part of the introduction.

Response: As suggested, the general objective and specific objectives of the study were expressed as “The purpose of this study is to find the genetic composition unique to pathogenic Listeria species, and to illustrate what is the role of them and which are the crucial functional gene clusters in their pathogenic lifestyle.”, which has been added in the revised manuscript (lines 79-82).

Minor suggestions:

Concern #1: Authors indicated that there are a total of 29 species of Listeria, however, there are a total of 30 identified Listeria spp. In this case, L. ilorinensis y L. swaminathanii have been discovered in 2022. Maybe one of them was discovered and included after authors consulted LPSN. Authors should correct that (Line 41) as well as in Line 50 they have to correct the sum of non-pathogenic Listeria spp. (28 in total).

Response: Thank you for providing the up-to-date information on Listeria spp. We have updated it in the revised manuscript (lines 43, 52 and 169).

Concern #2: Line 41: pH rather than PH.

Response: As suggested, the “PH” has been corrected to “pH” in the revised manuscript (line 43).

Concern #3: Table 1: The definition of the acronyms should be included at the end of the table.

Response: As suggested, we have checked all the tables of the manuscript, and the definition of the acronyms have been added and marked in the revised manuscript (Table 1, CDS, coding sequence; Table 2; ORFs, open reading frames).

Concern #4: Table 1 (Line 93) – Listeria is written without italic spelling. Names of microorganisms should be in italic. Please change that. This is a repeated mistake that is found in different parts of the manuscript:

“Line 160, Line 161, Line 183 (L. seeligeri), Line 238, Line 267… among others”. Revise the whole manuscript to correct that.

Response: As suggested, we have carefully checked the spelling of the bacterial name and corrected them with italic spelling in the revised manuscript (lines 45, 103, 185, 186, 208, 344, 383).

Concern #5: Line 147: Numbers that start after a point should be written with letters (i.e. 275)

Response: As suggested, we have checked and corrected those sentences in the revised manuscript.

Concern #6: Line 218: Should that not be Figure 2?

Response: Thanks for pointing out the error. It should be Figure 1B and should not be Figure 2. It has been corrected in the revised manuscript (line 324).

Concern #7: Line 343: “Deletion” with “d” in lowercase.

Response: As suggested, the spelling mistake has been corrected in the revised manuscript (line 405).

Concern #8: Bibliography requires a whole careful reading and correction. Names of the microorganisms are not in italic; some of the titles of the articles start with an uppercase in all the word while others everything is un lowercase. All of that requires correction.

Response: In response to this concern, we have carefully checked and corrected the spelling and grammar mistakes. All the corrections have been marked in the revised manuscript (lines 168, 284). Moreover, all the names of the microorganisms have been formatted with italic.

Round 2

Reviewer 1 Report

Dear authors,

Thank you very much for your comprehensive reply. All concerns and revisions have been addressed successfully and explained comprehensively. This valuable manuscript can be accepted for publishing in its present form.